# Prescribing Analgesics to Older People: A Challenge for GPs

**DOI:** 10.3390/ijerph17114017

**Published:** 2020-06-05

**Authors:** Ljiljana Trtica Majnarić, Thomas Wittlinger, Dunja Stolnik, František Babič, Zvonimir Bosnić, Stjepan Rudan

**Affiliations:** 1Department of Internal Medicine, Family Medicine and the History of Medicine, Faculty of Medicine, University Josip Juraj Strossmayer, 31000 Osijek, Croatia; ljiljana.majnaric@mefos.hr; 2Department of Public Health, Faculty of Dental Medicine, University Josip Juraj Strossmayer, 31000 Osijek, Croatia; zbosnic191@gmail.com (Z.B.); stjepan.rudan@gmail.com (S.R.); 3Department of Cardiology, Asklepios Hospital, 38642 Goslar, Germany; 4Family Medicine Practice, Health Center Osijek, 31000 Osijek, Croatia; dunja.stolnik@gmail.com; 5Department of Cybernetics and Artificial Intelligence, Faculty of Electrical Engineering and Informatics, Technical University of Košice, 04201 Košice, Slovak Republic; frantisek.babic@tuke.sk

**Keywords:** prescription analgesics, older adults, general practice, multimorbidity, musculoskeletal diseases, chronic pain, mental disorders

## Abstract

Background: Due to population aging, there is an increase in the prevalence of chronic diseases, and in particular musculoskeletal diseases. These trends are associated with an increased demand for prescription analgesics and an increased risk of polypharmacy and adverse medication reactions, which constitutes a challenge, especially for general practitioners (GPs), as the providers who are most responsible for the prescription policy. Objectives: To identify patterns of analgesics prescription for older people in the study area and explore associations between a long-term analgesic prescription and comorbidity patterns, as well as the prescription of psychotropic and other common medications in a continuous use. Methods: A retrospective study was conducted in 2015 in eastern Croatia. Patients were GP attenders ≥40 years old (*N* = 675), who were recruited during their appointments (consecutive patients). They were divided into two groups: those who have been continuously prescribed analgesics (*N* = 432) and those who have not (*N* = 243). Data from electronic health records were used to provide information about diagnoses of musculoskeletal and other chronic diseases, as well as prescription rates for analgesics and other medications. Exploratory methods and logistic regression models were used to analyse the data. Results: Analgesics have been continuously prescribed to 64% of the patients, mostly to those in the older age groups (50–79 years) and females, and they were indicated mainly for dorsalgia symptoms and arthrosis. Non-opioid analgesics were most common, with an increasing tendency to prescribe opioid analgesics to older patient groups aged 60–79 years. The study results indicate that there is a high rate of simultaneous prescription of analgesics and psychotropic medications, despite the intention of GPs to avoid prescribing psychotropic medications to patients who use any option with opioid analgesics. In general, receiving prescription analgesics does not exceed the prescription for chronic diseases over the rates that can be found in patients who do not receive prescription analgesics. Conclusion: Based on the analysis of comorbidities and parallel prescribing, the results of this study can improve GPs’ prescription and treatment strategies for musculoskeletal diseases and chronic pain conditions.

## 1. Background

The prevalence of musculoskeletal diseases is increasing worldwide due to population aging and the epidemic of obesity [1]. These conditions have been recognized as the leading cause of chronic non-cancer pain and physical disabilities [2]. Surveys on chronic pain in Europe have shown that there are significant differences between countries in terms of managing chronic pain and analgesics prescription policies [3,4]. The challenge of analgesics prescription mostly refers to general practitioners (GPs), as health care providers who bear the major responsibility for the provision of the prescription policy. An important fact that makes analgesics prescription challenging for GPs is that musculoskeletal diseases are particularly prevalent among older patients with multimorbidity [5]. This situation raises concerns about adverse medication reactions, due to polypharmacy and altered pharmacokinetics and pharmacodynamics in these patients. An additional important fact that contributes to the risk of adverse medication reactions in older people is that these medications are usually used on a long-term basis. For example, in older people, a long-term treatment with non-steroidal anti-inflammatory drugs (NSAIDs) is known to interfere with renal and cardiac function, or precipitate gastrointestinal (GI) tract disorders [6]. Another issue of concern is associated with the fact that musculoskeletal diseases provoke chronic pain, which is a cause of poor sleep [7]. Poor sleep, chronic pain and mental health disorders are interrelated disorders that are known to augment each other [8]. Together, this leads to a high demand for the prescription of both analgesics and medications with psychotropic actions.

In addition to classical analgesics, paracetamol and NSAIDs, opioids are now also available and they are considered as the preferred medication option for treating chronic non-cancer pain [9]. GPs in many European countries are now facing the new challenge of opioid misuse and overuse [10]. The chronic use of opioids in older people may lead to nervous system and psychiatric disorders [11]. The risk of adverse reactions involving opioid analgesics is particularly high in those older patients who simultaneously use anxiolytics or other psychotropic medications [12]. Research is being carried out in some European countries, with the aim of determining the size of the inappropriate analgesics prescription problem, as well as trends in the simultaneous prescription of analgesics and psychotropic medications [10,13,14]. Having recognized this problem, the European Pain Federation has released a position paper on appropriate opioid use in managing chronic pain [15]. However, specific guidelines that would help GPs to manage chronic pain in older members of the population have not been issued.

The aim of this study was to identify patterns of analgesics prescription in older people in the study area (eastern Croatia). A further aim was to identify patterns of comorbidities and medications prescription for mental health issues and other common chronic medical conditions, in older patients who have been continuously received prescription analgesics. The results are expected to improve GPs’ decision-making in this area’s prescription and treatment strategies, and more generally to inform their prescription policy.

## 2. Methods

### 2.1. Study Design, Setting and Patients

In 2015, a retrospective study using secondary data analysis was conducted in three PC practices located in two health centres in a wide area surrounding the town of Osijek (80,000 citizens), the administrative centre of eastern Croatia. In recent decades, this region has suffered the consequences of negative social processes, such as low economic growth, intensive emigration and population aging, all of which have influenced the high level of chronic diseases.

In Croatia, GPs have a gatekeeping role, which means that information on medication prescriptions is available in PC electronic health records (eHRs), for the majority of the population. Analgesics policy is strongly restrictive for over-the-counter (OTC) analgesics, so that prescription analgesics strongly reflect the real-life situation regarding analgesics consumption.

The study population comprised patients who were 40 years old and older. This age threshold marks the beginning of the period in life when chronic diseases are likely to emerge. The participants were GP attenders who were recruited at their appointments during a six-month study. This is a sufficiently long period that most older people who need medication for chronic diseases rotate. Overall, 675 patients were included in the study, of whom analgesics have been continuously prescribed to 432 patients, and not to the remaining 243 patients.

### 2.2. Data and Outcomes 

The study did not involve direct participation by the patients. The data on diagnoses of musculoskeletal and other common chronic diseases, as well as prescription analgesics and other prescription medications for treating common chronic diseases, were obtained from PC electronic health records (eHRs). The study was conducted according to the STROBE statement [16].

The prescribing of analgesics (and other medications) was considered long term if they had been prescribed for a continuous period of at least three months during the past year. The diagnoses of chronic diseases were recorded in the International Statistical Classification of Diseases and Related Health Problems, 10th revision (ICD-10). The medication group labelling was performed according to the WHO’s anatomical therapeutic chemical (ATC) classification system.

In this study, anxiolytics, antidepressants, antipsychotics and other medications for the nervous system, such as antiepileptics and hypnotics, were medications with psychotropic actions. In addition to paracetamol, non-opioid analgesics included a wide range of NSAIDs. The most frequently-prescribed opioid analgesic was tramadol, taken either alone or in combination with paracetamol, sold in a fixed-dose combination. The proportion of patients who received strong opioid analgesics was generally low. The most preferred among this group of analgesics were fentanyl, morphine, and oxycodone.

The aim was to identify differences between those patients who have been continuously prescribed analgesics and those who have not, in areas such as: (1) diagnoses of musculoskeletal and other diseases indicating pain; (2) diagnoses of psychiatric and other common chronic diseases; (3) patterns of prescription psychotropic medications and prescription medications for common chronic diseases; and (4) the rate of overlap between diagnoses of musculoskeletal and psychiatric diseases and the simultaneous prescription of analgesics and psychotropic medications.

### 2.3. Analytical Methods

Absolute frequencies and proportions were used to describe the numerical values, and the non-parametric Mann–Whitney test was used for comparison. The categorical variables were described using the median and interquartile range and tested using the Chi-square test or Fisher’s exact test (Table 1, Table 2 and Table 3). The age-related distribution (in ten-year age intervals) of patients receiving particular groups of prescription analgesics, as well as the rates of overlap between diagnoses of musculoskeletal and psychiatric diseases and simultaneous prescription of the respective groups of analgesics and psychotropic medications, were visualized graphically as bar diagrams (Figure 1 and Figure 2).

Multivariate analysis was performed using logistic regression (LR) models. This method was used to determine the probability of prescribing psychotropic medications in individuals who have received prescription analgesics, compared with those who have not, when adjusted for confounding factors (LR model 1) (Table 4 and Table 5), as well as the probability of prescribing medications for chronic diseases when analgesics have been continuously prescribed, compared with the situation in which they have not, and the factors influencing this prescription (LR model 2) (Table 6 and Table 7). For each model, we selected another data sample based on the results of univariate analysis and domain knowledge. To study relationships in LR models, we fitted several LR models and stepwise adjusted for covariates.

The 95% confidence interval (CI) was used to estimate the precision of the odds ratio (OR), with the bias-corrected and accelerated bootstrapping (BCa) containing a selected number of replications. The main advantage of the BCa interval is that it corrects for bias and skewness in the distribution of bootstrap estimates. The Akaike information criterion (AIC) was used to measure the quality of the model’s prediction performance [17]. The AIC estimates the quality of each model relative to each of the other models, based on the trade-off between goodness of fit and simplicity.

The analyses were performed using R, an open-source language and environment for statistical computing and graphics.

## 3. Results

The participants were mainly in the 56–76-year-old age range (median 65), and there were no significant differences between those who have been continuously prescribed analgesics and those who have not. The M to F ratio was 32.3% to 67.7%. Analgesics have been continuously prescribed to 64% of the participants. Among them, women participated with 71.3%. The most frequently prescribed were non-opioid analgesics (54.4%), while opioid and a combination of non-opioid and opioid analgesics had similar rates (23.6% and 21.9%, respectively) (Table 1).

The patients who have not been continuously receiving prescription analgesics have only occasionally received a prescription, mainly for injuries and fractures. By contrast, those patients who have been continuously receiving prescription analgesics had a wide range of musculoskeletal disease diagnoses, with the most prevalent being those from the “intervertebral disc disorders and other dorsopathies and dorsalgia symptoms” groups (M50–M54 in the ICD-10 labeling) and “arthrosis” (M15–M19 in the ICD-10 labeling). The second most common group of diagnoses indicating pain were neurologic disorders with a low participation rate (Table 2).

A high proportion of the participants (78.9%) had some diagnoses of psychiatric disease, with these diseases being more prevalent among those who have not received prescription analgesics (96.7%) than among those who have (68.9%). The most prevalent groups of psychiatric diseases were “neurotic, stress-related and somatoform disorders” (F40–F49 in the ICD-10 labeling) and “mood (affective) disorders” (F30–F39 in the ICD-10 labeling). Both groups were more prevalent among participants who had not received prescription analgesics compared with among those who have (71.2% vs. 55.8% and 18.9% vs. 12.7%) (Table 2).

A total of 73.2% of the participants had some type of common chronic disease, and there was no difference between the two patient groups. The most common comorbidities were diseases of the circulatory system (59.4%), and endocrine, nutritional and metabolic diseases (25.2%), which did not show differences between the two patient groups. Regarding the disease groups, a significant difference was only found for chronic diseases of the respiratory system (Table 2).

Fewer patients who have been prescribed analgesics were receiving psychotropic medications than those who have not (74.1% vs. 86.4%). Anxiolytics were the most common type of prescribed psychotropic medications, followed by antidepressants, whose prescription rates were lower for patients who have been prescribed analgesics than for those who have not (Table 3).

By contrast, medications for common chronic diseases have been prescribed at higher rates for patients with continuous analgesic prescriptions than for those without (79.4% vs. 71.2%). Regarding groups of medications, differences were found for medications for gastrointestinal, cardiovascular, and chronic respiratory diseases (Table 3).

Analgesics were prescribed to a total of 64% of patients who were 40 years old and over. These patients were mostly in the 50–79-year-old age group, which constitutes about half (47.8%) of all the patients, or three-quarters (74.2%) of those with continuous analgesics prescription. Prescribing was lower but still high in the very old patient group (80–89 years), compared with patients in the younger age group (40–49 years) (Figure 1).

In the younger patient group (40–59 years), prescription rates for non-opioid analgesics exceeded those of opioids and a combination of opioid and non-opioid analgesics. In the older patient group (60 years and over), prescription rates for these two groups of analgesics were more similar or inverse.

LR model 1 indicates that having been prescribed analgesics does not affect the probability of having been prescribed psychotropic medications (Table 5). After adjusting with covariates, some factors that were significant in the univariate analysis become non-significant (Table 4). The major factors associated with the probability of having been prescribed psychotropic medications include having been prescribed non-opioid analgesics, and having a diagnosis of dorsalgia symptoms and then arthrosis. 

After adjusting the model for the variable indicating diagnoses of mental diseases and other chronic diseases, the association between the variable indicating a 60–79 year age range patient group and the probability of having been prescribed psychotropic medications is lost. On the contrary, having been prescribed opioid or opioid and non-opioid analgesics diminishes (although not significantly) the probability of having been prescribed psychotropic medications. This association is also lost after entering into the LR model 1 variables indicating diagnoses of mental and other chronic diseases. 

Among the diagnoses of common mental health conditions, those indicating mood disorders but not those indicating neurotic, stress-related and somatoform, significantly contribute to the probability of having been prescribed psychotropic medications, albeit in a way that reduces this probability. This association still remains significant after adjusting for covariates, indicating the comorbidity context (number and types of chronic diseases).

Overall, LR model 2 (Table 7) indicates that having been prescribed analgesics—compared with not having done so—does not increase the probability of having been prescribed medications for chronic diseases, nor after adjusting for several covariates, such as those indicating age and gender, and those indicating prescribing for common chronic diseases including CV, chronic respiratory, and gastrointestinal diseases, or after adjusting for having been prescribed anxiolytics, antidepressants and antihypnotics. Having been prescribed for certain chronic diseases, including chronic respiratory and gastrointestinal diseases, increases this probability, whereas having been prescribed psychotropic medications reduces this probability. The significant association between gender (M) and having been prescribed medications for chronic diseases becomes non-significant after adjusting for the prescription of psychotropic medications and medications for chronic diseases (Table 7).

A large majority of patients from both patient groups—those receiving only non-opioid analgesics, and those receiving opioid analgesics or a combination of non-opioid and opioid analgesics—also receive prescription psychotropic medication, with an overlap ratio being somewhat favored in the “non-opioid patient group” (72% vs. 77%) (Figure 2a).

Many more patients with musculoskeletal diseases have had dual (M+F) rather than a single (M) diagnosis labeling (Figure 2b). The prescription rates of non-opioid analgesics vs. opioid analgesics and a combination of non-opioid and opioid analgesics are high and similar in the two patient groups (57% vs. 59%, respectively). A significant number of patients who do not have musculoskeletal diseases, and are diagnosed with some of the psychiatric diseases (F)—also in a great proportion (67.2%)—receive prescription analgesics, which are equally partitioned between non-opioid and opioid and a combination of non-opioid and opioid therapy groups (Figure 2b).

## 4. Discussion

### 4.1. Principal Findings, Comparison with Other Studies and Interpretation

In general, patients who receive analgesics use them on a long-term basis. These patients are mainly women 50–79 years old. Musculoskeletal diseases—notably dorsalgia symptoms, and arthrosis—are the reason for prescribing analgesics for most people of older age. This is in line with global statistics indicating musculoskeletal diseases as the major cause of chronic non-cancer pain, with lower back pain as the leading single cause [2]. If people have no problem with musculoskeletal diseases, they only occasionally express demand for prescription analgesics, for injuries and fractures. 

According to the latest report of the Pain Alliance Europe (2017), every fifth citizen in Europe suffers from chronic pain [15], although statements relating to the older population are scarce. The results of this study—which indicate that a high proportion (64%) of GP-attending adult patients (≥40 years) continuously use prescription analgesics—are therefore difficult to compare with those of other studies, due to differences in population characteristics and methodological issues such as the patient sampling procedure [16]. Since consecutive patients were used in the study, there might be a selection bias towards those with more comorbidities. 

The main known predisposing factors for musculoskeletal diseases and chronic pain experience are lower socioeconomic status, low physical activity and female gender [2]. Recent studies link chronic pain with frailty status in older women [18]. All of these mentioned factors are highlighted in this study: the domination of women among those receiving prescription analgesics, a socio-economically deprived region and the lack of the community-based physical activity and rehabilitation programs for older citizens. The rapid trend of population aging in this region in recent years combined with the neglect of older people may be the reason for the high prevalence of frail older women, and it can explain the high prescription rates of both analgesics and psychotropic medications. Specifically in this environment, traditional norms that keep women indoors and stereotypes like the notion that “it is normal for older women to be obese” may also contribute to the high level of musculoskeletal diseases and high demand for prescription analgesics. The fact that the prescription policy for OTC analgesics in Croatia is highly restrictive can also partly explain the high analgesics prescription rates. As indicated by this study, the pattern of prescription analgesics for older people in the study area is similar to that in most of Europe, with non-opioid analgesics being most common and an increasing tendency to prescribe opioid analgesics to older patient groups aged 60 years and older [4,19]. This pattern reflects the prescription strategy based on the European chronic pain management guidelines, which are known to promote weak opioids when first-line medications (paracetamol, NSAIDs or cyclo-oxygenase-2 inhibitors) are insufficiently effective, or if there is a risk of adverse reactions due to the long-term use of NSAIDs [15,19]. As a strategy to avoid the potential negative consequences of long-term NSAIDs and opioid analgesics consumption, the GPs in the study area also frequently use the benefits of the fixed marketing formula comprising paracetamol and a weak opioid—tramadol—or a combination of the two therapies, used at a lower dose. 

In contrast to the US and Canada, where opioid analgesics prescription rates are much higher than those in Europe, the European situation regarding the prescription of opioid analgesics is still not worrisome. In particular, the use of strong prescription analgesics remains low [15]. In this study, it was found to be sporadic. However, the concern remains that this favorable situation could suddenly change into the inappropriate prescription of opioid analgesics, as has recently been shown in the UK [10].

A large number of participants have some diagnoses of psychiatric diseases (78.9%) and receive psychotropic medications (78.5%), mostly for neurotic, stress-related and somatoform disorders, which are followed by mood disorders and behavioral (mostly sleep) disorders. These facts support the statement about older people living in the study area being burdened with existential issues and chronic diseases, and not having the ability to gain the benefits of non-pharmacological treatments. 

The incipient analysis indicated that fewer individuals receive prescription psychotropic medications among those who receive prescription analgesics (74.1%), than among those who do not (86.4%) (Table 3). We tested this hypothesis through a multivariate regression analysis (LR model 1, Table 4), which showed that having been prescribed analgesics does not significantly increase the probability of having been prescribed psychotropic medications, even after adjusting for many covariates. Nevertheless, this analysis has revealed many important factors that can influence the prescription of psychotropic medications—either positively or negatively—in older PC patients receiving prescribed analgesics. Knowledge on these factors may be useful when creating a prescription policy.

The main factor found to be significantly associated with the prescription of psychotropic medications is the prescription of non-opioid analgesics. This seems reasonable due to the high rates of simultaneous prescriptions (Figure 2a). 

Further analysis has revealed the complex interplay that may exist between the strategies used by the patients in their demands for prescription of these medications, and those used by the GPs when trying to respond to such demands. Although the rate of the simultaneous prescription shows an increasing tendency when the therapy with analgesics also includes the opioid component (77%) compared with when it does not (72%); the lower overall prescription rate of analgesics that include the opioid component is likely to neutralize this potentiating effect on the prescription of psychotropic medications (Table 2, Figure 2a).

In this regard, results of LR model 1 show that the significant positive effect of analgesics prescription on the prescription of psychotropic medications is limited to the 60–79-year age range patient group. In this patient group, the prescription of analgesics that include the opioid component shows an increasing tendency relative to the prescription of non-opioid analgesics (Figure 1).

However, the possible positive effect of this tendency on the prescription of psychotropic medications is counteracted with the strategy used by GPs, to avoid prescribing analgesics with the opioid component when prescribing psychotropic medications. This scenario is suggested by the results presented in Figure 2b. It can be seen that a dual diagnosing—a combination of diagnoses of musculoskeletal diseases and psychiatric diseases (“MF” column)—does not change the ratio between the prescription of analgesics that include the opioid component and non-opioid analgesics compared with this ratio in a situation with the single musculoskeletal disease diagnosis labeling (“M” column). The lacking predomination of prescribing analgesics with the opioid component in the “MF” patient group compared with the “M” patient group (“MF” vs. “M” ratio: 59% vs. 57%) is contrary to expectations. Namely, in the “MF” patient group, it is considered that the musculoskeletal pain is complicated by mental health symptoms due to the long-term pain duration. As suggested by the guidelines, these conditions are indications for the prescription of opioid or a combination of non-opioid and opioid analgesics [15]. A failure in the expected shift in the analgesics prescription patterns can be explained by an attempt by GPs to avoid prescribing analgesics that contain the opioid component in the presence of the “F” diagnosis.

On the contrary, the dominant expression of musculoskeletal health problems—as in the “M” patient group (Figure 2b)—does not exclude the prescription of psychotropic medications to serve as concomitant therapy for the health problems accompanying the intensive pain sensation, such as troubled sleep [7].

The GPs’ avoidance strategy, as suggested by the results of this study, can be multi-faceted. The first option may be to shift the prescription of analgesics with the opioid component to prescribing non-opioid analgesics (this option allows for the simultaneous prescription of psychotropic medications). The existence of this option in this study is indicated with the balanced rather than disproportionate prescription of non-opioid analgesics vs. opioid and a combination of non-opioid and opioid analgesics. The second option may be to avoid prescribing psychotropic medications simultaneously with analgesics with the opioid component. In this study, this option is suggested with the lower prescription rate of psychotropic medications demonstrated in the “MF” compared with the “M” patient group (87% vs. 97%) (Figure 2b). Alternatively, due to the sedation effects of these substances, therapy, including opioid analgesics, may diminish a demand of patients for prescribing psychotropic medications, despite the presence of prominent mental health symptoms [9].

The avoidance prescription strategy, as reconstructed from the results of this study, may explain the paradoxical finding, indicating that diagnoses of the common mental disorders in patients receiving the prescribed analgesics compared with those who do not have no effect—or have a significant negative effect—on the probability of receiving prescribed psychotropic medications (Table 5). This effect is more pronounced for the diagnoses of mood disorders (mostly including depression) compared with diagnoses of neurotic, stress-related, and somatoform disorders, and it can be explained by the stronger association between depression—rather than anxious disorders—and chronic musculoskeletal pain [20,21].

The results of LR model 1 indicating diagnoses of the common musculoskeletal diseases, arthrosis and dorsalgia related syndromes as being significantly—and independently of other factors—associated with the probability of having been prescribed psychotropic medications, thus supporting the evidence. In this respect, it is well known that psychological and cognitive factors have a mediating role in sustaining chronic pain, including lower back pain (dorsalgia syndrome) in particular [20]. In addition, in older patients with chronic musculoskeletal pain, depression may appear with the increased rates as a consequence of long-term treatment with opioid analgesics [22].

There is increasing awareness that the presence of mental health symptoms and chronic pain in people of older age and a demand for the prescription of analgesics and psychotropic medications should be considered more comprehensively in the context of individual patient psychologic characteristics and experience regarding chronic disease comorbidity patterns [23,24,25]. In this study, after the introduction into LR model 1 of the diagnoses of mental disorders and chronic diseases, the variable “age 60–79” is no longer associated with the prescription of psychotropic medications. According to Figure 1, after this age, a demand for the prescription of analgesics decreases in general, and for analgesics with the opioid component in particular. One of the reasons for this decreasing trend can be the shift in the dominant comorbidity patterns, which are known to change along with increasing age [5].

Consequently, mental health symptoms may appear in the context of other chronic diseases, rather than in the context of musculoskeletal disease-related chronic pain [23]. This scenario is suggested by the fact that a number of individuals with the diagnoses of psychiatric diseases (“F” column in Figure 2b), use analgesics regardless, whereas a number of these patients do not, although they are still receiving the prescribed psychotropic medications.

Based on the basic analyses showing that patients who have received prescription analgesics—compared with those who have not—have been prescribed more medications for chronic diseases (Table 3); despite the similar level of chronic diseases (Table 2), we tested the hypothesis that the prescription of analgesics increases the probability of prescription for chronic diseases (LR model 2, Table 5).

The results found were twofold. The hypothesis was not confirmed (ORs of adjusted LR models were about 1), which means that, overall, the prescription of analgesics does not exceed the prescription for chronic diseases over the rates that can be found in patients who do not receive prescription analgesics. Although prescriptions for some particular chronic diseases—such as chronic respiratory and gastrointestinal diseases—were shown to independently increase this probability, this reflects GPs’ analgesics prescription strategies.

In this respect, it is known that the prescription of protective agents for gastrointestinal tract disorders is used as concomitant therapy to the long-term prescription of NSAIDs [26]. An independent contributing effect of prescription for chronic respiratory diseases to the overall prescription for chronic diseases can be viewed in the context of the avoidance behavior of GPs when prescribing analgesics that contain the opioid component. This avoidance is due to the fear of side effects such as respiratory depression, which may in turn be followed by a switch to the prescription of non-opioid analgesics [27]. Alternatively, chronic respiratory diseases are associated with multiple comorbidities including arthrosis, which can increase the demand for prescribing non-opioid analgesics and other medications in general [28].

The significant reducing effect of prescribing the main groups of psychotropic medications—anxiolytics and antidepressants—on the overall prescription for chronic diseases can be viewed in the context of the switch that usually occurs with an increase in age (Figure 1), from the dominant prescription as concomitant therapy to the prescription of analgesics, to the dominant prescription for mental symptoms associated with multiple chronic diseases [29]. It may be the case that the beneficial effect of psychotropic medications on anxious symptoms and sleep problems—and consequently on the quality of life—reduces the patient adherence to treatment for chronic diseases [30]. There is a need to increase GPs’ awareness on this issue due the contra-productive long-term effects of such a prescription policy on patient outcomes [31].

Taken together, research on the exact analgesic prescription patterns—especially for older patients in a PC setting who are characterized with comorbidities and polypharmacy (parallel prescribing)—is scarce [14,32]. Evidence is also scarce on the rates of simultaneous prescription of opioid analgesics and psychotropic medications, specifically for older population groups [33]. In addition, the side effects of the long-term prescription of both opioid analgesics and psychotropic medications on patient outcomes—especially in conditions associated with multiple comorbidities—are poorly known [31,34]. Although it can be expected that GPs—as this study also illustrates—take care regarding appropriate prescriptions and potential deleterious drug-drug interactions when prescribing analgesics, it is not possible to avoid all potentially contra-indicated situations [14]. Indeed, this is especially difficult, due to the need for a long-term treatment [6]. Since the research process is difficult to conduct in conditions of comorbidities, there is insufficient information on the benefit-risk ratio of particular prescription patterns to inform guidelines and enable the implementation of individualized prescription strategies for older people with comorbidities [35].

In the current situation, when no effective implementation strategy has been established for managing lower back pain, such analyses, as demonstrated in this study, could help GPs in recognizing older patients with painful conditions who are candidates for the misuse and overuse of both analgesics and psychotropic medications [27,36]. As indicated by the results of this study (Table 1), patients who continuously receive prescription analgesics are also frequent visitors to GPs [37]. The implementation of the simple alerting tool in eHRs for automatically screening such patients might be useful for focusing GPs’ attention on problematic situations for medication prescriptions.

As recognized by other authors and suggested by the results of this study, there is a need to implement, in practice, more integrated strategies for managing chronic pain and mental health issues that would be able to reduce the demand for medications. These strategies should focus on cognitive-behavioral methods, education about the importance of maintaining physical fitness, and organizing physical recreation programs for older citizens living in the local community [38].

### 4.2. Strengths and Limitations

This study is a contribution to reports from European countries about patterns in prescribing analgesics to older adults. Its main strength lies in providing a comprehensive review that considers parallel prescribing and diagnosis labeling, indicating the prescription and comorbidity patterns of patients to whom analgesics have been prescribed, compared with those to whom they have not.

It is emphasized that by using routine data from eHRs and multivariate methods for data analysis, it is possible to draw under the surface of the complexity of comorbidities, as the environment in which prescription for older patients in PC takes place. The study has revealed many details of such prescription process and identified the “hot spots” in this process that should be further evaluated in future research, necessarily using qualitative research methods.

This is also the main limitation of this study. Since it relies on automated health data, information was not available on patients’ cognitive processes nor their mental and physical functioning, which in turn would hold strong importance for understanding the pain perception and the demand for analgesics and/or psychotropic substances. For this reason, this study could also not adequately address the balance between the appropriateness and risks of treatment with analgesics in older adults.

A further shortcoming is related to the fact that a limited sample size was used for analysis due to difficulties in manual data collection, and the selection procedure was to take consecutive patients, which may produce some degree of bias when extrapolating to the general population. Although analgesics in Croatia are prescription-controlled medications, OTC analgesic distribution and sharing among layers have not been taken into account.

### 4.3. Conclusions and Directions for Future Research

In order to reduce analgesics consumption and the level of polypharmacy in older patients in a GP setting, a radical change is needed in the therapy of chronic musculoskeletal pain, which should be oriented towards the more significant implementation of non-pharmacological treatments. These should include methods such as cognitive-behavioral, cognitive coping, and appraisal techniques (positive coping), as well as community-based physical recreation and rehabilitation programs. Educating patients about the importance of physical activity should become a part of regular prevention programs organized by public health institutions and GPs. More content on the role of musculoskeletal diseases in generating comorbidities, as well as how these diseases contribute to exacerbating disability and frailty in older people, should be added to medical students’ learning curricula and GPs’ postgraduate education. Research into the effect of prescribing analgesics to older people in the context of multimorbidity and polypharmacy should be intensified, particularly by including qualitative research approaches, which would likely shed more light on the problem of the intersection between the perception of pain and the mental and global health status.

## Figures and Tables

**Figure 1 ijerph-17-04017-f001:**
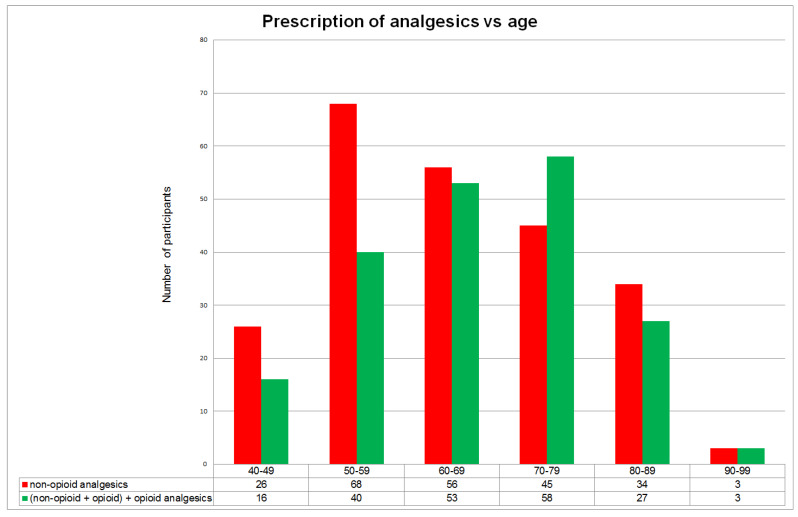
Age-related distribution (ten-year intervals) of continuous analgesics prescription, regarding non-opioid vs. opioid and combined non-opioid+opioid analgesics.

**Figure 2 ijerph-17-04017-f002:**
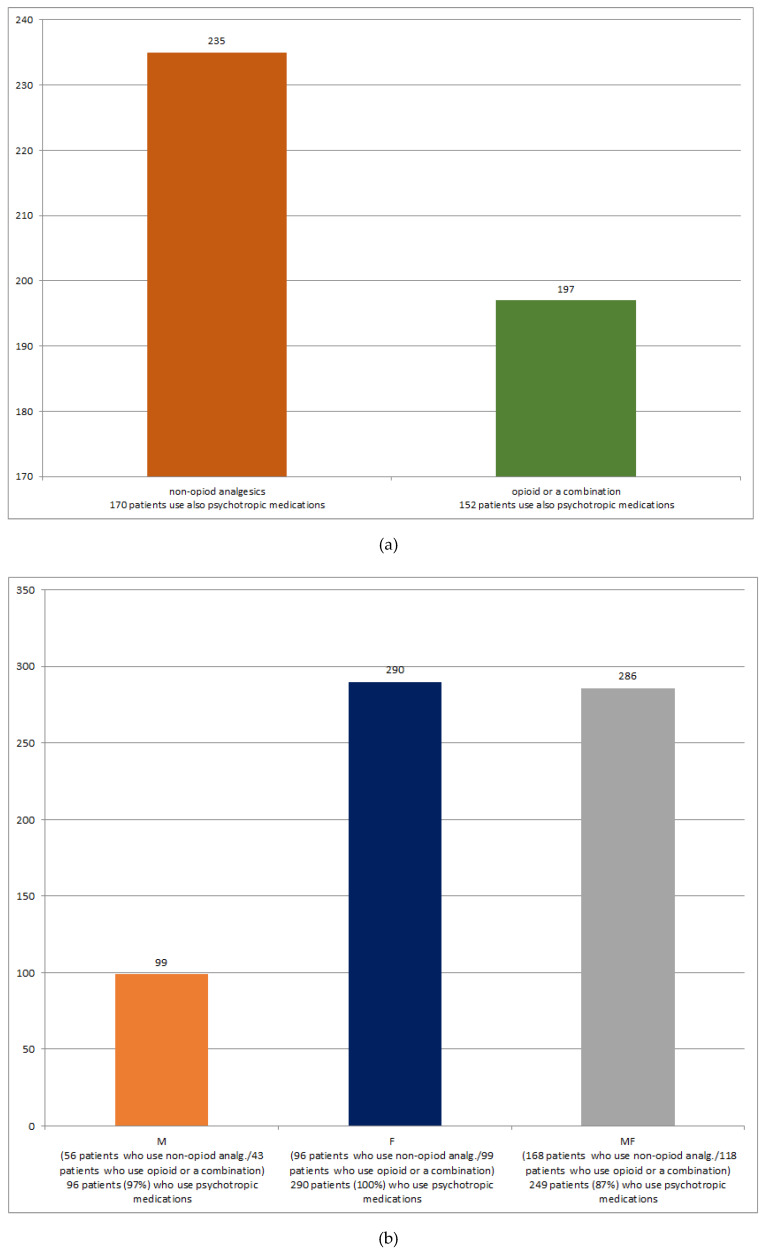
Rate of simultaneous prescription of a respective group of analgesics and psychotropic medications (**a**) and the rate of overlap between diagnoses of musculoskeletal (M) and psychiatric (F) diseases (**b**).

**Table 1 ijerph-17-04017-t001:** Analgesics prescription rates and basic characteristics of individuals receiving prescription analgesics (I.) and those who have not (II.)

Numerical Variables	Median (IQR)		Mann–Whitney Test *p*-Value	Median (IQR) Total	
	I.	II.			
Age Number of annual visits	66 (57–75)	64 (55–76)	0.45	65 (56–76)	
	7 (3–14)	4 (1–8)	<0.001	6 (2–12)	
	**Categorical Variables**	**Number of Patients (%)**		**Chi-Square Test** ***p*-Value**	**Number of Patients (%)** **Total**
		**Yes**	**No**		
	Analgesics	Total 432 (100.0)			
	Non-opioid	235 (54.4)			
	Dual prescription non-opioid and opioid	95 (21.9)			
	Opioid	102 (23.6)			
	Gender				
	Female	308 (71.3)	149 (61.3)	0.008	457 (67.7)
	Male	124 (28.7)	94 (38.7)		218 (32.3)
	Total	432 (100.0)	243 (100.0)		675 (100.0)

IQR—interquartile range, 0.05 significance level.

**Table 2 ijerph-17-04017-t002:** Differences in diagnoses (comorbidity patterns) between individuals with prescribed analgesics (I.) and those without (II.)

Categorical Variables	Number of Patients (%)		Chi-Square Test *p*-Value	Number of Patients (%) Total
	I.	II.		
I. Diagnoses indicating pain (All)	432 (100)	23 (9.5)	<0.001	455 (67.4)
(a) Musculoskeletal diseases (All)	418 (96.7)	15 (6.2)	<0.001	439 (65.0)
Arthrosis	115 (26.6)	3 (1.2)	<0.001	118 (17.5)
Intervertebral disc disorders and dorsalgia symptoms	296 (68.5)	7 (2.9)	<0.001	303 (44.9)
Inflammatory polyarthropathies	51 (12.0)	0	<0.001	52 (7.7)
Other	75 (17.4)	7 (2.9)	<0.001	42 (6.2)
(b) Neurologic conditions	12 (2.8)	0	0.005 *	12 (1.8)
(c) Injuries and fractures	2 (0.5)	8 (3.3)	0.006 *	10 (1.5)
II. Diagnoses of psychiatric diseases (All)	297 (68.9)	235 (96.7)	<0.001	532 (78.9)
(a) Neurotic, stress-related and somatoform	241 (55.8)	173 (71.2)	<0.001	414 (61.3)
(b) Mood disorders	55 (12.7)	46 (18.9)	0.03	101 (15.0)
(c) Organic, including symptomatic, mental	15 (3.5)	13 (5.3)	0.24	28 (4.1)
(d) Others	58 (13.4)	68 (28.0)	<0.001	126 (18.7)
III. Diagnoses of other chronic diseases (All)	325 (75.2)	169 (69.5)	0.11	494 (73.2)
(a) Endocrine, nutritional or metabolic	113 (26.2)	57 (23.5)	0.44	170 (25.2)
(b) Diseases of the circulatory system	267 (61.8)	134 (55.1)	0.09	401 (59.4)
(c) Diseases of the respiratory system	32 (7.4)	6 (2.5)	0.008	38 (5.6)
(d) Diseases of the digestive system	79 (18.3)	39 (16.0)	0.46	118 (17.5)
(e) Diseases of the genitourinary system: urinary system, pelvis, genitals and breasts	16 (3.7)	11 (4.5)	0.60	27 (4.0)

0.05 significance level, * Fisher’s exact test.

**Table 3 ijerph-17-04017-t003:** Differences in prescription patterns between individuals with (I.) and without (II.) prescribed analgesics.

	Number of Patients (%)		Chi-Square Test *p*-Value	Number of Patients (%) Total
	I.	II.		
I. Psychotropic medications (All)	320 (74.1)	210 (86.4)	<0.001	530 (78.5)
(a) Anxiolytics	277 (64.1)	212 (87.2)	<0.001	489 (72.4)
(b) Antidepressants	58 (13.4)	81 (33.3)	<0.001	139 (20.6)
(c) Antipsychotics	13 (3.0)	41 (16.9)	<0.001	54 (0.1)
(d) Hypnotics	27 (6.2)	27 (11.1)	0.04	54 (8.0)
(e) Others	69 (16.0)	43 (17.7)	1.00	112 (16.6)
II. Medications for chronic diseases (All)	343 (79.4)	173 (71.2)	0.02	516 (76.4)
(a) Medications for blood diseases, inflammation, and gout	93 (21.5)	45 (18.5)	0.35	138 (20.4)
(b) Medications for endocrine diseases	105 (24.3)	46 (18.9)	0.11	151 (22.4)
(c) Medications for gastrointestinal diseases	111 (25.7)	40 (16.5)	0.006	151 (22.4)
(d) Medications for cardiovascular system and the kidneys	270 (62.5)	134 (55.1)	<0.001	513 (76.0)
(e) Medications for chronic respiratory diseases	33 (7.6)	3 (1.2)	<0.001	36 (5.3)

0.05 significance level.

**Table 4 ijerph-17-04017-t004:** Univariate analyses. Probability of having been prescribed psychotropic medications in patients receiving analgesics compared with those who have not, and factors associated with this probability.

Variable	Univariate Analyses (Pearson’s Chi-Squared Test)
Psychotropic medications	0.64
Non-opioid analgesics	<0.001
Opioid+(opioid+non-opioid) analgesics	<0.001
Age (≤59, 60–79, ≥80 years)	0.24
Sex	0.008
Arthrosis	<0.001
Intervertebral disc disorders and dorsalgia symptoms	<0.001
Inflammatory polyarthrosis	<0.001
Other musculoskeletal diseases	<0.001
Neurotic, stress-related, somatoform disorders	<0.001
Mood disorders	0.01
Behavioral (sleep) disorders	0.07
Organic (cognitive) disorders	0.24
Circulatory disease	0.14
Chronic respiratory diseases	0.01
Other chronic diseases	0.16
Several diagnoses of chronic diseases	0.42

0.05 significance level.

**Table 5 ijerph-17-04017-t005:** Multivariate analysis. Probability of having been prescribed psychotropic medications in patients receiving analgesics compared with those who have not, and factors associated with this probability.

Variable	Multivariate Analysis (Logistic Regression)			
	Model 1 OR_crude_ (95% CI)	Model 2 OR_adj._ (95% CI)	Model 3 OR_adj._ (95% CI)	Model 4 OR_adj_ (95% CI)
Psychotropic medications = yes	0.52	0.70	0.60	0.44
	(0.34–0.78)	(0.43–1.13)	(0.28–1.29)	(0.19–0.98)
	(0.44–1.41)	(0.40–1.31)	(0.21–2.16)	(0.12–1.82)
Non-opioid analgesics = yes **	529.28 *	623.75 *	607.49 *	678.12 *
	(100.41–2 790.04)	(118.88–3367.96)	(104.95–3516.45)	(94.60–4860.74)
	(128–8 594.5)	(177.7–11919.7)	(147.5–16900.4)	(97.9–94918.5)
Opioid+(opioid+non-opioid) analgesics = yes **	0.42	0.38	0.29	0.23
	(0.04–3.90)	(0.04–3.90)	(0.03–3.40)	(0.02–2.81)
	(0.0–101.2)	(0.0–88.3)	(0.0–171.5)	(0.12–299.4)
Age 60–79		2.30	1.39	1.78
		(1.38–3.82) *	(0.61–3.13)	(0.74–4.60)
		(1.39–4.58)	(0.51–4.21)	(0.50–8.2)
Age 80		1.32	1.30	1.05
		(0.96–2.50)	(0.47–3.55)	(0.35–3.16)
		(0.64–2.85)	(0.43–3.65)	(0.27–4.00)
Sex M		0.48	0.66	0.59
		(0.31–0.76)	(0.33–1.31)	(0.27–1.29)
		(0.27–0.82)	(0.25–1.51)	(0.23–1.86)
Arthrosis = yes **			13.33 *	55.4 *
			(5.04–35.8)	(17.07–179.9)
			(1.94–118.4)	(4.49–360.8)
Intervertebral disc disorders and dorsalgia symptoms = yes **			62.26 *	149.90 *
			(29.8–130.4) *	(60.01–373.96)
			(18.0–156.5)	(39.9–365.3)
Inflammatory polyarthrosis = yes				1.93
				(0.41–9.06)
				(0.04–51.32)
Other musculoskeletal diseases = yes				0.0008
				(0.00008–0.009)
				(0.0–0.29)
Neurotic, stress-related, somatoform = yes			0.83	0.37
			(0.70–1.73)	(0.18–0.77) *
			(0.21–1.62)	(0.15–1.31)
Mood disorders = yes			0.35	0.10
			(0.14–0.84) *	(0.03–0.29) *
			(0.06–0.85)	(0.03–0.50)
Behavioral (sleep) disorders = yes			0.34	0.53
			(0.14–0.86)	(0.19–1.42)
			(0.11–1.25)	(0.12–2.03)
Organic (cognitive) disorders =yes			1.66	0.90
			(0.48–5.77)	(0.23–3.58)
			(0.11–5.35)	(0.05–7.90)
Circulatory disease = yes			0.52	0.46
			(0.21–1.26)	(0.17–1.24)
			(0.18–1.56)	(0.12–1.78)
Chronic respiratory diseases = yes			4.22	2.14
			(1.13–15.75)	(0.49–9.38)
			(0.47–15.36)	(0.11–13.14)
Other chronic diseases = yes			1.60	1.76
			(0.58–4.40)	(0.59–5.22)
			(0.50–5.57)	(0.38–9.76)
Diagnoses of chronic diseases (1–3)				3.41
				(0.94–12.34)
				(0.36–15.0)

0.05 significance level. AIC (Model 1) = 420.54, AIC (Model 2) = 412.16, AIC (Model 3) = 234.91, AIC (Model 4) = 222.64 *—parameter with *p*-value <0.05 in respective LR model. Intervals marked in red—bias-corrected and accelerated bootstrap CI with 1000 replications **—BCa intervals used extreme quantiles within 1000 replications. We tested 5000 replications.

**Table 6 ijerph-17-04017-t006:** Univariate analyses. Probability of having been prescribed medications for chronic diseases in patients receiving analgesics compared with those who have not, and factors associated with this probability.

Variable	Univariate Analyses (Pearson’s Chi-Squared Test)
Medications for chronic diseases	0.74
Gender	0.01
Age: <60 years, >= 60 years	0.14
Anxyolitiycs	<0.001
Hypnotics	0.04
Antidepressants	<0.001
Medications for CV disease	0.10
Medications for chronic respiratory diseases	<0.001
Medications for gastrointestinal diseases	0.02

0.05 significance level.

**Table 7 ijerph-17-04017-t007:** Multivariate analysis. Probability of having been prescribed medications for chronic diseases in patients receiving analgesics compared with those who have not, and factors associated with this probability.

Variable	Multivariate Analysis (Logistic Regression)		
	Model 1 OR_crude_ (95% CI)	Model 2 OR_adj._ (95% CI)	Model 2 OR_adj._ (95% CI)
Medications for chronic diseases	1.02	1.08	1.06
	(0.74–1.41)	(0.77–1.54)	(0.75–1.51)
	(0.70–1.53)	(0.70–1.66)	(0.69–1.62)
Gender M	0.65	0.60	0.58
	(0.49–0.86) *	(0.48–0.82)	(0.42–0.79)
	(0.47–0.92)	(0.42–0.84)	(0.39–0.83)
Age: <60 years	1.26	1.31	1.30
	(0.95–1.68)	(0.96–1.78) *	(0.96–1.78) *
	(0.92–1.77)	(0.90–1.95)	(0.89–2.01)
Anxiolytics = yes		0.25	0.24
		(0.17–0.36) *	(0.16–0.35) *
		(0.17–0.39)	(0.16–0.39)
Hypnotics = yes		0.54	0.49
		(0.17–0.92)	(0.28–0.84) *
		(0.25–1.25)	(0.22–1.07)
Antidepressants = yes		0.32	0.33
		(0.23–0.46) *	(0.24–0.47) *
		(0.21–0.53)	(0.22–0.55)
Medications for CV disease = yes			1.07
			(0.80–1.44)
			(0.75–1.54)
Medications for chronic respiratory diseases = yes			7.16
			(2.53–20.32) *
			(2.62–81.59)
Medications for gastrointestinal diseases = yes			1.74
			(1.22–2.49) *
			(1.06–2.76)

0.05 significance level. AIC (Model 1) = 883.4, AIC (Model 2) = 808.02, AIC (Model 3) = 791.2 *—parameter with *p*-value < 0.05 in respective LR model. Intervals marked in red—bias-corrected and accelerated bootstrap CI with 1000 replications.

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
