# Peer review of "Prescribing Analgesics to Older People: A Challenge for GPs"

_ijerph, 2020, doi:10.3390/ijerph17114017_

Round 1

Reviewer 1 Report

The authors have improved their manuscript however I would stress that the validation of the confidence intervals of the odds ratio must be done after bootstrapping with more than 90% of the runs converging. This validation will confirm significant associations and update the results/discussion accordingly.

Reviewer 2 Report

The manuscript entitled “Prescribing analgesics to older people: a challenge for the GPs” by Majnarić and coworkers investigated the patterns of analgesics prescription for older people to explore associations between a long-term analgesic prescription and comorbidity patterns, as well the prescription of psychotropic and other common medications in a continuous use.

Their findings show that analgesics have been continuously prescribed mostly to older people and females and were indicated mainly for dorsalgia symptoms and arthrosis. The study indicated that there is a high rate of simultaneous prescription of analgesics and psychotropic medications, despite an intention of GPs to avoid prescription of psychotropic medications in patients who use any option with opioid analgesics. Taking into account, analgesics prescriptions do not exceed prescription for chronic diseases over the rates that can be found in patients who do not receive prescription analgesics.

This study addresses an important problem among aging populations such as multimorbidity and polypharmacy and provides new evidence that can improve the GPs in prescription and treatment strategies for musculoskeletal diseases and chronic pain conditions. 

On the whole, this study seems well conducted, methods are sound, data are convincing and the conclusions are in general supported by the data. However, there are a few issues, which need to be addressed:

  • In the methods section, in the aim of the study point 2 is mentioned twice
  • In Table 2, III d no parenthesis
  • Table 4 and 5 are not readable, the authors should improve the readability of the tables
  • In Figure 2, in column F the number of patients administered non-opioid and opioid drugs should be provided
  • References should be ordered

Round 2

Reviewer 1 Report

Most of the suggestions were taken into consideration.

This manuscript is a resubmission of an earlier submission. The following is a list of the peer review reports and author responses from that submission.

Round 1

Reviewer 1 Report

This manuscript adds to the growing literature in Europe regarding analgesic prescribing, focusing on an older population in Croatia. The retrospective study used electronic medical records to look for correlations between analgesic prescribing and other comorbidities, and with prescriptions of other medications. While this sort of analysis is possibly valuable from a general perspective - general practitioners need to know that their patients might be taking multiple medications - the analysis suffers from trumping up what amount to spurious correlations and there is a fair bit of p-hacking going on. The discussion must be edited before this manuscript is acceptable for publication. 

Specific comments (in order of the text): 
line 107-108: what are "opioids with safe pharmacological profiles," and how do fentanyl, morphine, and oxycodone count? This might be a simple grammatical oversight. 

Tables: the use of markers (e.g., thorn, dagger, asterisk) and italics is confusing. Recommend the use of superscript a,b to distinguish between different statistical tests and * for significance values below a certain threshold.

Tables: it does not make much sense to run statistical analyses between every group. For example, why would it matter whether there is a statistically significant difference between patients who received an analgesic to treat "other dorsopathies" and those who didn't? What hypotheses were the authors testing that they felt this was necessary? It is unclear why this would be of importance to GP's. 

Line 148: "...than among those who have been not" seems to be incorrect; per Table 2, 68.9% of patients receiving analgesics have a diagnosis of psychiatric disease. 

Line 155: "borderline significance" is a misnomer, as two values are either mathematically significantly different, or they are not. Perhaps there is a trend toward significance, but it is not clear why that would be of importance here. 

Lines 157-158: per the data in Table 3, 74.1% patients who received prescription analgesics also received a prescription for psychotropic medications, which is less than the 86.4% of patients who did not receive prescription analgesics. This contradicts the language here. This discrepancy needs to be addressed. 

Table 3: this needs to be cleaned up; for example, antiepileptics are not antidepressants, though they are listed under the same heading. Similarly, hypnotics and medications for the nervous system are not antipsychotics. 

Figure 1: the text talks about 2 groups of patients: those taking an opioid+combination analgesic and those only taking a non-opioid analgesic (see line 183). This top part of the figure might be better shown with the opioid+combo in a single bar (with different colors to distinguish the two), and non-opioid only. The color scheme in the bottom figure should be changed; otherwise, the reader will think that the bars mean the same thing in the two parts of the figure. Y-axes must be labeled. Instead of % of total population, the top might be better served as raw numbers of patients, as is shown in the bottom. 

Table 4: I am not sure why linear regression models were run. What were the dependent and independent variables, and why did the authors feel there might be linear relationships between them? Again, where is the hypothesis to be tested?

Line 212, Discussion in general: there is lots of causation language where there shouldn't be. For example, lines 221-222 suggest that prescription of psychotropic medications influences analgesic prescriptions, which was not measured in this study. In fact, causation is unable to be determined from a retrospective study such as this. Another example is line 223, which attempts to explain why patients had high rates of psychiatric drug prescribing, which, again, is unsupported by the data here. 

Lines 227-233: while I agree with the overall premise that the elderly might be at greater risk for complications, ADR's, etc., there is no evidence given here that these patients, indeed, suffered as a result of polypharmacy. This also generally limits the overall impact of the study. "They took multiple medications, the GP was aware of it, and they were fine. No problem." 

Line 238-239: the data indicate that patients who receive prescription analgesics continuously are, in fact, LESS likely to take other medications (Table 3, and see commentary above). 

Lines 242-243: again, DDI's and ADR's are certainly possible when prescribing multiple medications, though the data here do not show that this is a problem. Since this is discussed in the "strengths" section, there needs to be some actual evidence in the paper that this was studied. Did the authors see examples where psychotropic medications negatively influenced the activity of analgesics? If not, this strength is not supported.

A general comment: did the authors note that the same GP, or different GPs, were prescribing all medications for all patients, or is this a situation where the authors feel that GPs need to take care to review their patients' records thoroughly? 

Line 284: the comment that "controlled substances are not under strict control" in the USA and Canada needs re-wording, since this makes no sense. Perhaps the authors mean to say that opioid analgesic prescribing rates are much higher in the USA and Canada than in Europe?

Line 310: Table 3 still says that the % of patients who receive medications for psychotropic medications is lower than those who do not. 

Lines 309-328: there is more p-hacking here and spurious correlations drawn from linear regression models. None of this discussion relates to the data collected. 

Reviewer 2 Report

Please! Perform appropriate univariate analyses followed by multivariate analyses as should be done with confirmation of confidence intervals with bootstrapping (see an example of (Corchia A, Wynckel A, Journet J, Moussi Frances J, Skandrani N, Lautrette A, Zafrani L, Lewandowski E, Reboul P, Vrigneaud L, Djerada Z, Rieu P. Metformin-related lactic acidosis with acute kidney injury: results of a French observational multicenter study. Clin Toxicol (Phila). 2020 May;58(5):375-382).

Please discuss the impact of these prescribing habits and their risk. Discuss the direction to be taken in terms of the benefit-risk balance.
